

# Analyzing the relationship between self-efficacy and impulsivity in amateur soccer referees

José López-Aguilar[1], Rafael Burgueño[2], Alfonso Castillo-Rodriguez[1] and Wanesa Onetti-Onetti[3]

[1] Departamento de Educación Física y Deportiva, Universidad de Granada, Granada, España
[2] Health Research Centre, University of Almeria, Almeria, Spain
[3] Universidad Internacional de la Rioja, Logroño, España

## ABSTRACT

Soccer referees (SRs) are exposed to stressful situations during the competition that can affect decision-making, could be affected by impulsivity during the competition and therefore, require optimal psychological skills. The objective of this study was to ascertain and analyze the relationship between levels of impulsivity and self-efficacy of amateur SRs in the senior category. A total of 21 Spanish SRs participated in this study (age $23.57 \pm 2.40$ years and $7.81 \pm 2.58$ seasons of experience). Self-efficacy data were collected with the REFS questionnaire between 48 and 72 h before the competition. The impulsivity data were collected using the UPPS-P questionnaire 60 min before the start the competition. The results indicate that SRs with higher self-efficacy have lower levels of impulsivity, specifically in the dimensions of negative urgency ($p < 0.01$), positive urgency ($p < 0.05$), lack of premeditation ($p < 0.001$), and lack of perseverance ($p < 0.001$), as well as lower global impulsivity ($p < 0.01$). However, the SRs with the highest self-efficacy also obtained higher levels in the sensation seeking dimension ($p < 0.05$). In conclusion, the main finding of this study showed that self-efficacy is influenced by SR impulsivity prior to competition. These novel facts allow us to discover aspects related to decision-making in refereeing that can be trained to reach optimal levels.

## INTRODUCTION

Soccer is one of most practiced sports in the world (*Pérez-Gómez et al., 2020*), with more than 265 million people who often practice it (*Castillo-Rodríguez et al., 2012*) and more than 5 million soccer referees (SRs) who perform their work as a match official (*FIFA, 2007; Malaguti et al., 2019*). SRs are athletes with the essential function of making the official competition develop correctly, making decisions based on the regulations (*Soriano Gillué et al., 2018*). Over time and considering the evolution that soccer has been undergoing, the physical requirements and performance of players and SRs have become greater (*Rampinini et al., 2007; Rebolé et al., 2016*). In addition, as SRs must be close to the action, they must acquire a mental strength that differentiates them from non-athletes

Corresponding author
Alfonso Castillo-Rodriguez,
acastillo@ugr.es

(*Guillén & Laborde, 2014*) developing specific psychological skills, such as self-confidence, concentration, motivation (*Castillo-Rodríguez, López-Aguilar & Alonso-Arbiol, 2021*; *Ramírez et al., 2006*) or self-efficacy (*Guillén & Feltz, 2011*; *Guillén et al., 2019*). These psychological skills allow SRs to make decisions in extreme situations (public, possible promotions, relegations, etc.) within a context of permanent competition stress (*Soriano Gillué et al., 2018*) and, consequently, achieve excellent refereeing (*Garcés de los Fayos & Vives, 2003*; *Giske, Hausen & Johansen, 2016*; *Weinberg & Richardson, 1990*).

Firstly, one of the constructs that plays a fundamental role in sport performance in general and football referee performance in particular is self-efficacy (*Aragão e Pina et al., 2021*). *Bandura (1997)* defined self-efficacy as the strength of a person's conviction to successfully adopt a behavior required to achieve a desired outcome. As self-efficacy is inherent to the target activity to be undertaken (*Bandura, 1997*), referee's self-efficacy constitutes a specific type of self-efficacy in accordance with *Bandura (1997)*. Particularly, referee's self-efficacy is conceptualized as the belief that (s)he has in being able to successfully carry out their work (*Guillén & Feltz, 2011*). Specifically, *Myers et al. (2012)* operationalized referee's self-efficacy into four dimensions: (a) game knowledge, refers to the confidence that SRs have in knowing their sport/regulations; (b) decision-making, refers to the confidence and ability of the SRs to make decisions during the game; (c) pressure, refers to the referee's confidence in not being influenced by the pressure of the match; and (d) communication, by which we mean the referee's ability to communicate effectively. In the general sport context, athletes' self-efficacy was positively associated with performance and negatively with anxiety and stress (*Guillén et al., 2019*). In the specific context of refereeing, the study carried out on handball referees stands out (*Diotaiuti et al., 2017*) with the results showing that the referees of the highest category and experience obtained higher levels of self-efficacy. Likewise, in a study of basketball referees, it was observed that the older ones had greater self-efficacy (*Karacam & Adiguzel, 2019*). Moreover, in SRs from a higher category, age and experience reveal higher levels of self-efficacy, also obtaining moderate positive relationships of self-efficacy with the variables listed above (*López-Aguilar et al., 2021b*).

One of the most common psychological factors in athletes, including soccer referees, is impulsivity (*Liebel, Edwards & Broglio, 2021*). Previous research has held the premise that impulsivity is a related yet different construct that indicates deficits in planning, perseverance, and inhibitory control (*Liebel, Edwards & Broglio, 2021*). In particular, impulsivity is broadly conceptualized through the multidimensional approach by *Whiteside et al. (2005)*, which theoretically differentiates between five dimensions: (a) positive urgency refers to the propensity to experience strong reactions to possible positive situations; (b) negative urgency refers to the propensity to experience strong reactions to possible negative situations; (c) lack of premeditation is the tendency not to think about the consequences of a certain action before taking it; (d) lack of perseverance, considered as the inability to maintain concentration on a specific job or task that can be long-lasting, boring or that presents some difficulty for the person; and (e) sensation seeking, which is linked to the tendency to engage in exciting activities associated with risky behaviors. Previous studies conducted in the general sport setting have consistently

shown that impulsivity was positively associated with higher injuries rates and lower sport participation (*Weishaar et al., 2021*), motivationally significant, affective conditions (*Holfeder et al., 2020*), and poor sport performance (*Siekanska & Wojtowicz, 2020*). In the specific field of refereeing, *López-Aguilar et al. (2021a)* study found that impulsivity was influenced by age and experience, showing that those soccer referees with greater experience and age recorded lower levels of impulsivity.

Thus, after considering the importance of the two constructs in sport and more specifically in refereeing, it was decided to study the possible relationship that both may have according to the following justification based on scientific evidence. Self-efficacy is negatively related to stress and anxiety (*Guillén et al., 2019*). Consequently, the higher the self-efficacy of the SR, the lower the levels of stress, which means that they will not have alterations in visual and motor perception (*Tornero-Aguilera & Clemente-Suárez, 2018*). This will allow them to take more information from the environment in order to make good decisions (*Castillo-Rodríguez et al., 2018*) which, consequently, will not be impulsive. For these reasons, guided by *Bandura's (1997)* theory, we could put forward the previously unpublished hypothesis that SRs with high self-efficacy will have lower levels of impulsivity. Bandura's self-efficacy theory maintains the idea that when someone is perceived as effective, they have greater control over their behavior because they have a strong and positive judgment of their abilities to do the behavior (desired) or avoid the behavior (undesired), which leads us to say that they have the skills of self-control (*Bandura, 1997*). This theory could be affected by impulsivity, studied as a personality trait that acts without evaluating the possible effects (*Whiteside et al., 2005*).

Despite the importance attributed to referees' self-efficacy in developing optimal performance (*Aragão e Pina et al., 2021*), little is known about the relationship between soccer referees' self-efficacy and their impulsivity, a construct that is also poorly addressed in this field. Because of the existing research gap, there is a need for more research on these two constructs and their relationship in order to test their effect on sport performance in general and football referee performance in particular. To the best of our knowledge, no studies have been found to analyze this hypothetical relationship. Therefore, this study could help SRs, physical trainers and sport psychologists to incorporate training interventions that will allow modular behaviors to be able to make decisions with less impulsivity. The objective of this study is to ascertain and analyze the relationship between different levels of self-efficacy and impulsivity in a sample of amateur SRs in the senior category.

## MATERIALS AND METHODS

### Participants

The participating sample included 21 soccer referees (19 men and two women) aged between 18 and 27 years old ($M = 23.57$, $SD = 2.40$). Of the total of referees, five belonged to the national category, while 16 belonged to the Honor Division or regional category. They self-reported an experience of from 3 to 14 seasons ($M = 7.81$, $SD = 2.58$). A non-probabilistic sampling method was followed to select and recruit participants. After

contacting the Andalusia Committee, they, in turn, communicated it to the SRs. In total, the sample was made up of 24 soccer referees. However, despite the complete and clear instructions, the responses of three referees had to be excluded because the impulsivity test was performed 1 day before (24 h before), instead of 1 h before the competition. This implied a possible alteration in the response and its exclusion from the analysis was determined. All participants volunteered to carry out the study and were informed of the procedures, objectives, methodology, benefits and possible risks involved. This study was approved by the Ethics Committee of the University of Granada, Spain (471/CEIH/2018).

## Instruments

### Self-efficacy in soccer referees

The Spanish version (*Guillén et al., 2019*) of the Referee Self-efficacy Scale (*Myers et al., 2012*) was used to judge soccer referees' perceptions of self-efficacy. The instrument is proceeded by the statement "When you perform as a referee, what level of confidence do you have in your capacity to…" and followed by 13 items to measure game knowledge (through three items, *e.g.*, "understand the basic strategy of the game?"), decision making (through three items, *e.g.*, "make quick decisions?"), pressure (through three items, *e.g.*, "be uninfluenced by pressure from players?"), and communication (through four items, *e.g.*, "communicate effectively with players?"). Items are responded to using a five-point Likert-type scale ranging from 1 (very low) to 5 (very high). In line with prior research (López-Aguilar et al., 2021), a mean score for referees' self-efficacy was computed by averaging the scores of the four factors comprising it.

### Impulsivity in soccer referees

The Spanish short version (*Cándido et al., 2012*) of the UPPS Impulsive Behavior Scale (*Whiteside et al., 2005*) was used to assess soccer referees' perceptions of impulsivity. The instrument is headed by the statement: "Please, indicate your degree of agreement with each of the following sentences" and includes 20 items that, grouped into five per factor, measure negative urgency (*e.g.*, "When I feel bad, I will often do things I later regret in order to make myself feel better now"), positive urgency (*e.g.*, "I tend to lose control when I am in a great mood"), lack of premeditation (*e.g.*, "My thinking is usually careful and purposeful"), lack of perseverance (*e.g.*, "Unfinished tasks really bother me"), and sensation seeking (*e.g.*, "I welcome new and exciting experiences and sensations, even if they are a little frightening and unconventional"). Items are rated on a four-point Likert-type scale ranging from 1 (strongly disagree) to 4 (strongly agree). Consistent with previous studies (*Guillén et al., 2019*), a mean score for referees' impulsivity was estimated by taking the average scores of each factor comprising it.

## Procedure

This research was a descriptive cross-sectional study of a single sample of each questionnaire that took place between the months of January and March 2020. First,

the SRs were located to explain the objectives, methodology and research protocols. In addition, all subjects provided us with their written informed consent, and were told that when they were going to referee a match in their category, they would be contacted between 48 and 72 h before the match so that they could complete the REFS questionnaire on self-efficacy without being influenced by the competition. Subsequently, after informing and notifying them, they would carry out the UPPS-P impulsivity questionnaire 60 min before the games at the sports facilities. Regarding the exclusion criteria, it was established that the referees should not have suffered serious injuries in the last 6 months that could affect them in the normal development of the matches. This criterion was added taking into account the emotional and psychological fluctuations in general that can affect the athlete in moments of absence due to injury (*Zafra et al., 2011*).

## Data analysis

Prior to main analyses, a screening of data was conducted, where neither univariate outliers (*i.e.*, Z scores over 3) nor multivariate outliers (*i.e.*, Mahalanobis $d^2$ at $p < 0.001$) were identified (*Field, 2017*). First, the normality assumption was analyzed by skewness and kurtosis coefficients, which are representative of a normal distribution when standardized values are as high as 1.96 (*Field, 2017*). Second, reliability of every target variable was examined by Cronbach's alpha ($\alpha$), which is acceptable when values are above 0.70 (*Viladrich, Angulo-Brunet & Doval, 2017*). Third, mean scores and standard deviation for each variable under study were computed to inform descriptive statistics. Four, Pearson's bivariate correlations among all variables under study were estimated. Five, a multivariate analysis of covariance (MANCOVA) test was run to analyze the relationship between different soccer referees' levels of self-efficacy and the five dimensions of impulsivity. For this analysis and following previous research (*Sicilia, Ferriz & Sáenz-Álvarez, 2013*), two groups in terms of self-efficacy were created by considering the mid-point of the Likert-type scale of self-efficacy. The first group (*i.e.*, higher self-efficacy group) consisted of 13 soccer referees who scored higher than the mid-point of the measurement scale in self-efficacy, while the second group (*i.e.*, lower self-efficacy group) included eight soccer referees who scored lower than the mid-point of the scale in self-efficacy. Given that age, category, and experience influenced self-efficacy and impulsivity among referees (*López-Aguilar et al., 2021a, 2021b*), age, category and experience were, respectively, introduced as covariates for the MANCOVA test. Consistent with *Field (2017)*, partial eta squared ($\eta_P^2$) was calculated as an effect-size measure. A small, medium and large effect size is reflected in values equal to 0.10, 0.25 and 0.50, respectively (*Richardson, 2011*). Previously to MANCOVA, Box's test was run to assess the homogeneity of covariances. The results supported the assumption of homogeneity of covariances (Box's $M = 23.64$, $F_{(15,\ 871.09)} = 1.06$, $p = 0.389$), proposing the use of Wilk's lambda ($\lambda$) as a test statistic (*Field, 2017*). The level of statistical significance was set at $p < 0.05$ (*Field, 2017*). Data were statistically analyzed with the Statistical Package for the Social Sciences (version 25.00, IBM® SPSS®, Chicago, IL, USA).

**Table 1 Descriptive statistics and reliability coefficients for the study variables.**

| | Min | Max | M(SD) | $\gamma_1$ | $\gamma_2$ | α | 1 | 2 | 3 | 4 | 5 | 6 | 7 | 8 | 9 |
|---|---|---|---|---|---|---|---|---|---|---|---|---|---|---|---|
| 1. Age | 18 | 27 | 23.57(2.40) | –0.67 | 0.37 | – | – | 0.17* | 0.54*** | 0.25** | 0.31*** | 0.25** | –0.36*** | 0.06 | –0.01 |
| 2. Category | 1 | 2 | 1.67(0.48) | –0.76 | –1.58 | – | | – | 0.03 | 0.36*** | 0.23** | 0.37*** | –0.09 | –0.28** | 0.34*** |
| 3. Experience | 1 | 10 | 3.95(2.40) | 0.83 | 0.68 | – | | | – | 0.55*** | 0.81*** | 0.30*** | –0.33*** | 0.07 | 0.35*** |
| 4. Self-efficacy | 1 | 5 | 4.54(0.38) | –0.40 | –1.13 | 0.88 | | | | – | –0.22** | –0.12* | 0.06 | –0.02 | 0.03 |
| 5. Negative urgency | 1 | 4 | 2.41(0.91) | –0.37 | 0.08 | 0.88 | | | | | – | 0.75*** | –0.39*** | –0.10 | 0.57*** |
| 6. Positive urgency | 1 | 4 | 2.35(0.83) | 0.16 | 0.15 | 0.86 | | | | | | – | –0.36*** | –0.32*** | 0.65*** |
| 7. Lack of premeditation | 1 | 4 | 2.01(0.76) | 0.43 | –0.44 | 0.71 | | | | | | | – | 0.60*** | –0.26** |
| 8. Lack of perseverance | 1 | 4 | 2.11(0.70) | 0.10 | –0.86 | 0.81 | | | | | | | | – | –0.18* |
| 9. Sensation seeking | 1 | 04 | 2.73(0.87) | –0.26 | –0.79 | 0.88 | | | | | | | | | – |

**Notes:**
Min, Minimum value; Max, Maximum value; α, Cronbach's alpha.
*** $p < 0.001$.
** $p < 0.01$.
* $p < 0.05$.

## RESULTS

### Descriptive statistics and reliability coefficients for the study variables

Table 1 shows standardized scores between –0.40 and 0.43 for skewness and between –1.13 and 0.15 for kurtosis, providing evidence in support of a normal data distribution. Furthermore, suitable reliability values were found with Cronbach's alpha ranging from 0.71 to 0.88. On the other hand, there were negative correlations between self-efficacy and negative and positive urgency. Alternatively, nonsignificant correlations were found between self-efficacy and the three remaining dimensions of impulsivity. In addition, there were mean scores above the mid-point of the measurement scale for every variable under study.

### Mean differences in soccer referees' levels of impulsivity according to self-efficacy

MANCOVA tests revealed, after controlling for age (Wilk's $\lambda = 0.66$, $F_{(5, 12)} = 1.23$, $p = 0.356$, $\eta_P^2 = 0.33$), category (Wilk's $\lambda = 0.76$, $F_{(5, 12)} = 0.76$, $p = 0.593$, $\eta_P^2 = 0.24$) and experience (Wilk's $\lambda = 0.61$, $F_{(5, 12)} = 1.55$, $p = 0.246$, $\eta_P^2 = 0.39$), statistically significant multivariate effects for the two self-efficacy groups regarding the dimensions comprising impulsivity (Wilk's $\lambda = 0.18$, $F_{(5, 12)} = 19.97$, $p < 0.001$, $\eta_P^2 = 0.82$).

Table 2 shows that while the higher self-efficacy group obtained significantly greater mean scores in sensation seeking, the lower self-efficacy group scored significantly higher in negative urgency, lack of premeditation and lack of perseverance. Likewise, impulsivity was significantly higher in the lower self-efficacy group than the higher self-efficacy group.

## DISCUSSION

The objective of this study was to ascertain and analyze the relationship between impulsivity and self-efficacy of amateur SRs in the senior category. The results obtained

**Table 2 Mean differences by self-efficacy in soccer referees' levels of impulsivity.**

| | Higher self-efficacy group | Lower self-efficacy group | | | |
|---|---|---|---|---|---|
| | $M(SD)$ | $M(SD)$ | $F_{(5,12)}$ | $p$-value | $\eta_P^2$ |
| Negative urgency | 2.06(0.91) | 2.98(0.59) | 14.47 | 0.002 | 0.48 |
| Positive urgency | 2.15(0.78) | 2.67(0.85) | 4.27 | 0.055 | 0.21 |
| Lack of premeditation | 1.57(0.48) | 2.72(0.57) | 20.96 | <0.001 | 0.57 |
| Lack of perseverance | 1.69(0.46) | 2.79(0.42) | 21.81 | <0.001 | 0.58 |
| Sensation seeking | 3.13(0.62) | 2.06(0.83) | 7.46 | 0.015 | 0.32 |
| Impulsivity | 2.12(0.40) | 2.64(0.37) | 14.56 | 0.002 | 0.48 |

show that SRs with higher self-efficacy recorded lower impulsivity scores in general, with the sensation seeking dimension being higher in SRs with higher self-efficacy, possibly because, although they take more risk in their decision-making, they have the certainty that they are correct, thanks also to their experience. Thus, we can confirm the hypothesis that SRs with higher levels of self-efficacy will have lower levels of impulsivity. In this respect, it should be noted, justifying the results obtained in the correlation of this manuscript, that the SRs with more experience were those with the highest self-efficacy, confirming the results obtained in the studies by *Diotaiuti et al. (2017)* in handball referees, the study by *López-Aguilar et al. (2021b)* in soccer referees and *Karacam & Adiguzel (2019)* who demonstrated this claim in basketball referees. We highlight that the studies mentioned above also used the REFS questionnaire. In the present study, the MANCOVA test was calculated using age, category, and experience as covariates, to avoid the influence of these metrics on impulsivity and self-efficacy. In the literature on personality traits, studies that analyzed the changes due to age and experience that implied a decrease/increase in the levels of some variables, established a cut-off point close to 30 years, which was when a degree of experience of greater than 10 years was consolidated (*McCrae et al., 1999*; *Srivastava et al., 2003*).

Low levels of self-efficacy were related to greater stress and anxiety (*Guillén et al., 2019*), which could cause burnout in SRs and trigger the abandonment of sport practice. Moreover, *Tornero-Aguilera, Robles-Pérez & Clemente-Suárez (2017)* indicated that optimal self-efficacy could result in less impulsive decision making due to the absence or control of stress over time and alleviation of negative influence on the perception of the environment, body, time, cognition, and memory. So, this conclusion has been demonstrated in this study.

Various authors indicated the importance of carrying out training and psychological intervention programs that allow referees to acquire knowledge and mechanisms to face difficult situations during refereeing (*Ramírez et al., 2006*; *González-Oya & Dosil, 2007*; *Alonso-Arbiol et al., 2005*), and they may be especially important in novice and young referees, who, after verbal or physical attacks, could be led to abandoning the practice of refereeing (*Alonso-Arbiol et al., 2005*). In addition, the most experienced SRs could establish or increase these skills, which would translate into an improvement in their refereeing work (*Guillén, 2003*; *Ramírez et al., 2006*), evidenced in category promotions.

To develop the psychological skills of referees we could replicate the 8-week psychological intervention program based on self-talk conducted on young athletes, proposed in the study by *Walter, Nikoleizig & Alfermann, 2019*. It was demonstrated how this intervention program produced a decrease in somatic anxiety and an increase in self-confidence and self-efficacy among other variables (*Walter, Nikoleizig & Alfermann, 2019*) which could benefit SR performance.

## Practical implications

The importance of this study lies in providing the scientific community with a novel investigation on the fact of relating the constructs of self-efficacy and impulsivity in athletes and more specifically in SRs. It thus contributes to the knowledge of the SR personality and to establishing relationships between constructs that could be related. Taking into account that impulsivity, a variable related to the athlete's personality, is closely related to decision-making (*Mirzaei, Nikbakhsh & Sharififar, 2013*), this study could provide scientific evidence so that SRs can train and modulate their decisions and responses. These responses, which according to the study by *López-Aguilar et al. (2021a)* are similar to those of soccer players with a defensive role, who make more cautious decisions mainly due to the damage caused by an error, compared to players with an offensive role (*Castillo-Rodríguez et al., 2018*). Furthermore, self-efficacy is a construct that has been previously studied in SRs. The positive or negative perception is demonstrated as the SRs increase their experience in the competition, and the influence of stress on it; however, it has not been studied whether a personality trait such as impulsivity would affect said self-efficacy. This is the first study that shows that SRs with higher self-efficacy have a lower level of impulsivity.

## Limitations

There are several limitations to this study. First, the adoption of the non-probabilistic sampling strategy to recruit and select participants does not allow us to generalize the results to the whole population and, therefore, they should be interpreted with caution. Specifically, the participating sample that was accessed in the competitive period was from the national category and its previous one, called "Division of Honor", but it was not possible to choose an elite SR. Thus, a larger sample in higher categories could optimize the results obtained in this study. To carry out an in-depth analysis of the relation between self-efficacy and impulsivity, future research should consider the category, age and age range of each category. Second, it would be interesting to be able to compare these "cold" responses, *i.e.*, in basal states 48 h after the competition and before 48 h prior to the next competition, with the precompetitive or "hot" responses that could be collected 30 min before the competition, or in the 15-min break between one half of the match and the other. These responses could also provide information on the fluctuation of said constructs in the same SRs. It would, indeed, be interesting for future research to be able to consider the contextual factors of the match, *e.g.*, yellow and red cards, number of spectators, goals scored by the visiting team, etc., and analyze the studied psychological responses of impulsivity and self-confidence. Third, although the use of a cross-sectional

design in this research has permitted us to explore the possible relationships between different levels of self-efficacy and impulsivity self-reported in SRs, cause-effect relationships between both variables could not be established. Therefore, longitudinal and experimental research is additionally required to tackle the in-depth examination of the relationship between self-efficacy and impulsivity in SRs. Four, the use of only self-reported measures of self-efficacy and impulsivity in SRs. Thus, there is a need to utilize complementary observational instruments in order to provide a better understanding of the effects of self-efficacy on impulsivity in SRs.

## CONCLUSIONS

The main findings of the study showed that SRs with higher self-efficacy had lower levels of impulsivity prior to the competition, occurring both in one-dimensional impulsivity and in all dimensions except in the sensation seeking variable, accompanied by a large effect size. These novel facts allow one to discover aspects related to decision-making in SRs that can be trained to reach optimal levels. This study could help SRs, physical trainers, and sport psychologists to incorporate interventions in training that allow modulating behaviors in order to make decisions with less impulsivity.

### Funding

This work was supported by the Precompetitive Research Projects program for Young Researchers of the Own Plan 2020, of the University of Granada: PPJIA2020.04. This study has been financed by project FEDER/Junta de Andalucía-Consejería de Transformación Económica, Industria, Conocimiento y Universidades P20_00194, of the R+D+i project aid program. Aid for R+D+i, within the scope of the Andalusian Research, Development and Innovation Plan (PAIDI 2020) of the Junta de Andalucía (Spain). Rafael Burgueño is supported by a Margarita Salas postdoctoral fellowship (grant number: RR_A_2021_02) from the Spanish Ministry of Universities. The funders had no role in study design, data collection and analysis, decision to publish, or preparation of the manuscript.

### Grant Disclosures

The following grant information was disclosed by the authors:
University of Granada: PPJIA2020.04.
FEDER/Junta de Andalucía-Consejería de Transformación Económica, Industria, Conocimiento y Universidades: P20_00194.
Junta de Andalucía.
Spanish Ministry of Universities: RR_A_2021_02.

### Competing Interests

The authors declare that they have no competing interests.

## Author Contributions

- José López-Aguilar performed the experiments, authored or reviewed drafts of the paper, and approved the final draft.
- Rafael Burgueño conceived and designed the experiments, analyzed the data, prepared figures and/or tables, and approved the final draft.
- Alfonso Castillo-Rodriguez conceived and designed the experiments, performed the experiments, authored or reviewed drafts of the paper, and approved the final draft.
- Wanesa Onetti-Onetti conceived and designed the experiments, authored or reviewed drafts of the paper, and approved the final draft.

## Human Ethics

The following information was supplied relating to ethical approvals (*i.e.*, approving body and any reference numbers):

This study was approved by the Ethics Committee of the University of Granada (Ethical Application Ref: 471/CEIH/2018).

## Data Availability

The raw measurements are available in the Supplemental File.

## Supplemental Information

Supplemental information for this article can be found online at http://dx.doi.org/10.7717/peerj.13058#supplemental-information.

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
