# Peer review of "Analyzing the relationship between self-efficacy and impulsivity in amateur soccer referees"

_PeerJ, doi:10.7717/peerj.13058_

## Round 0.1 · original submission · Major Revisions

Three reviewers have provided substantial comments and suggestions. It requires major revision before I made a decision. Please revise your manuscript accordingly and highlight the change parts with color.

Reviewer 1 ·

Basic reporting

The article called "Analyzing the relationship between self-efficacy and impulsivity in soccer referees" studies psychological aspects developed in soccer referees in competitive period. It is about a theme that fits well with PeerJ journal.

The article is clear, has novel and English-language references and has relevant results for the fulfillment of the hypotheses.

However, with regard to tables and figures, I would like to indicate (not sure this is an internal PeerJ Journal standard), that "0" should be inserted. instead of ".".

Experimental design

Further explanation is required in the section on statistical analysis of effect size (partial eta square). I recommend that the authors indicate the level of significance at the end of the section. Please add the statistical methods used with your reference.

Finally, I request an increase in the discussion, where the results obtained are explained in more detail (compared to other referees from other sports for example).

Validity of the findings

no comment

Additional comments

This is an unpublished original study because impulsivity in soccer referees has never been studied before. It offers important results in order to increase efficiency and performance in a competitive period. Congratulations.

Reviewer 2 ·

Basic reporting

no comment

Experimental design

no comment

Validity of the findings

no comment

Additional comments

Title & Abstract
1. In title the researcher needed to mention participants so that the reader could have a better idea subjects from which part of the world they reading about. The abstract needs improvement, and it does not meet the requirements for publication in its original form. The results of the abstract need to present some supporting statistics, such as soccer referees level. Regarding the inappropriate presentation of p-values, it is recommended to change to P <0.05, P <0.01, P <0.001. The conclusions in the abstract do not present the conclusions that can be drawn from this work. It merely summarizes what was done. In addition, it is necessary to add the significance of this research and its contribution to this field.

Introduction
2. In my opinion, the introduction is far too long. Make it more concise, and much shorter. This is not a complicated study, and the introduction should reflect that. Why did you choose to study the associations between self-efficacy and impulsivity in soccer referees? How does self-efficacy affect impulsivity? Do you want to consider the moderating effects of gender and age? Is there a difference between ordinary people and soccer referees? The relationship between them is based on what theory, and what their mechanism is, it needs to be further clarified. Clearly identify the gaps in the literature. Is there a specific gap your study is filling? Again, is this gap specific to soccer referees or can it be generalized to other referees?The hypothesis of this research needs to be clear.

Results
3. Why is there no correlation analysis? Further analyze the moderating effects of gender and age on self-efficacy and impulsivity in soccer referees, and report it.

Discussion
4. Please start the discussion by stating the main results, without interpretations in light of other research (ie "This study's results further support those theories or models). After reading the discussion, I as a reader am not sure what the contribution of this paper is to the greater literature of self-efficacy and impulsivity in soccer referees. It isn't highlighted which gaps in knowledge this research targets, and whether it has succeeded in closing the gap, or at least moving it in that direction. Can any of these results be practically applied? The limited part is not detailed enough, for example, longitudinal research can be used to further verify the results of this study, and more parts need to be considered.

Conclusions
5. It is recommended to add the significance of this research and its contribution to this field.

Annotated reviews are not available for download in order to protect the identity of reviewers who chose to remain anonymous.

Reviewer 3 ·

Basic reporting

The authors conducted and reported on a study about the relationship between self-efficacy and impulsivity in soccer referees. The authors indicated that soccer referee is exposed to stressful situations during the competition that can affect decision-making, being able to be more or less impulsive during the competition and therefore requires optimal psychological skills. I thought this is an interesting and very timely topic. Nevertheless, in its current state, I have some major concerns with the manuscript that limit its overall contribution to the literature in referees’ psychological skills building in stressful situations during the competition.
1. The main issue in the introduction is that the contributions of the study have not been clearly articulated. More specifically, the authors did not clearly identify the research gap in the existing research literature. That said, I cannot realize that understanding the relationship between self-efficacy and impulsivity is an important issue for the soccer referee. For example, the authors mentioned that these SRs need to acquire specific psychological skills, such as self-confidence, concentration, motivation, or self-efficacy. However, why did this study want to focus on SRs' self-efficacy? I suggest you should have a well-developed story to explain why SRs' self-efficacy is most important for SRs and provide the arguments to demonstrate the relationship you want to analyze.
2. On lines 54-97, I thought the structure of the introduction seems strange for the reader. As the authors argued that this study is aimed to understand the relationship between self-efficacy and impulsivity for SRs. That is, you should first explain why self-efficacy plays a key role for the SRs. Then, introduce the variable of impulsivity, and the influence of impulsivity for these SRs. Lastly, based on the theory or references to argue why SRs with high self-efficacy will have lower levels of impulsivity.
3. On lines 93-95, although the authors mentioned that SRs with greater self-efficacy will have higher levels of self-confidence and will perform better in matches since they will not be influenced by high levels of stress and anxiety that may impair their decision-making. However, this section lacked a specific theory and arguments to support your hypothesis. Accordingly, the authors need to be clarified within the introduction to improve the study rationale and the manuscript overall.

Experimental design

In the Materials & Methods part, there are some suggestions for the authors below.
1. On lines 112-113, the authors explain that a non-probabilistic sampling method was adopted in this study. And the participants were 21 soccer referees aged between 18 and 27 years old. I suggest the authors have to provide more detailed information regarding how to choose the samples. As the reference you provide on lines 70-72, impulsivity was influenced by age and experience. That is, only focusing on the younger SRs might not provide comprehensive results and insights for the readers.
2. On line 115, this study was approved by the Ethics Committee of the University of XXXXXXX. What is the University of XXXXXXX?
3. On lines 133-134, the authors mentioned that the scale of impulsivity in soccer references grouped into 4 factors. However, as you demonstrated later, this scale was grouped into 5 factors. Please check if this is correct or not.
4. On lines 153-155, the authors provide the exclusion criteria for the readers. However, I cannot understand why the exclusion criteria of data were referees had not suffered serious injuries in the last 6 months.
5. I suggest the authors provide how much data you got from 21 SRs. Because I cannot get any information regarding how many times you got the data from each SR or the amount of data from each SR. This is an important issue about the validity of your findings.

Validity of the findings

In the Results and Discussion part, there are some suggestions for the authors below.
1. MANCOVA test was applied to analyze test the hypothesis in this study. On lines 196-198, the authors mentioned that SRs’ age, category, and experience were control variables. I suggest you should provide the rationale to explain why these variables need to be controlled.
2. On lines 153-155, the authors mentioned that the objective of this study was to know and analyze the relationship between levels of impulsivity and self-efficacy of amateur SRs in the senior category. The sentence about the relationship between levels of impulsivity and self-efficacy seems not to correspond with your research topic.
3. On lines 243-244, according to the results, the authors demonstrated that SRs can control their responses and decision-making through greater self-efficacy. But this is a cross-sectional study to test the relationship between self-efficacy and impulsivity in soccer referees. I thought the authors' argument might be over-explain the result.
4. In addition, it seems to lack more details about the insights and contributions of this result. More specifically, I suggest the authors can provide practical intervention for SRs on how to improve their self-efficacy.

I hope these comments are useful when the authors further develop this paper.

---

## Round 0.2 · Major Revisions

Two reviewers gave lots of comments. You need to revise it accordingly. I will make my final decision as it meets the reviewers' standards.

Reviewer 2 ·

Basic reporting

no comments.

Experimental design

no comments.

Validity of the findings

no comments.

Additional comments

The author has responded to some questions, but some revisions are still required to meet the requirements of publication.

Materials & Methods
-Why is the sample size recovery 100%, an incredible result?
-Instruments needs to add some information, such as Cronbach’s alpha and Confirmatory factor analysis (CFA).
- Add Instruments factor loading of each item scope?

Results
-Max. and min. values should be included in Table 1.

Discussion
-The implications of this study need further clarification, and it is best to write a paragraph about the implications of this study.
-The Limitations section is obviously too small, and an additional paragraph describing the limitations of this study is needed.
- The manuscript requires English-speaking authors to revise the grammar of the full text.

Reviewer 3 ·

Basic reporting

The authors have modified the manuscripts to respond to my suggestions. I congratulate the authors on their efforts. However, I expressed concerns with the arguments developed regarding the relationship between self-efficacy and impulsivity. In addition, the sentence about the relationship between levels of impulsivity and self-efficacy in the abstract and discussion was not consistent with your research topic. Below are more detailed comments.

1. On lines 98-100, although the authors mentioned they further synthesize the introduction according to the scientific results to explain SRs with high self-efficacy will have lower levels of impulsivity. However, I still can't find what theory and what mechanism can support your hypothesis, it needs to be further clarified.

2. The research topic is analyzing the relationship between self-efficacy and impulsivity in amateur soccer referees. However, the sentence in the abstract and discussion (on lines 227-228) demonstrated that this study was to analyze the relationship between levels of impulsivity and self-efficacy. I suggest the authors have to display consistency in the sentence.

I hope the comments are useful for the authors to further develop this study.

Experimental design

The authors have responded to my suggestions.

Validity of the findings

The authors have responded to my suggestions.

---

## Round 0.3 · accepted · Accept

Congratulations. I have just received the final comments from three reviewers. They are satisfied with your revision.

Reviewer 1 ·

Basic reporting

The authors have provided answers to both my initial questions and the 2 fellow reviewers. I accept the current version of the manuscript.

Experimental design

The authors have provided answers to both my initial questions and the 2 fellow reviewers. I accept the current version of the manuscript.

Validity of the findings

The authors have provided answers to both my initial questions and the 2 fellow reviewers. I accept the current version of the manuscript.

Additional comments

no comments

Reviewer 2 ·

Basic reporting

no comments.

Experimental design

no comments.

Validity of the findings

no comments.

Additional comments

The manuscript has been revised to the level of publication by the authors.

Reviewer 3 ·

Basic reporting

Thank you for the revision. I found it adequately addressed my concerns. The manuscript is well-written, and the modifications have proved effective in improving the manuscript.

Experimental design

no comment

Validity of the findings

no comment

Additional comments

no comment